# Natural Killer cells demonstrate distinct eQTL and transcriptome-wide disease associations, highlighting their role in autoimmunity

James J. Gilchrist [1,2,3,7✉], Seiko Makino[3,7], Vivek Naranbhai[3,7], Piyush K. Sharma[2,4], Surya Koturan[2,4], Orion Tong [2,4], Chelsea A. Taylor [2,4], Robert A. Watson[2,4], Alba Verge de los Aires[2,4], Rosalin Cooper[2,4], Evelyn Lau[3], Sara Danielli[3], Dan Hameiri-Bowen [3], Wanseon Lee[3], Esther Ng[3], Justin Whalley [3], Julian C. Knight [3,5,7✉] & Benjamin P. Fairfax [2,4,6,7✉]

Natural Killer cells are innate lymphocytes with central roles in immunosurveillance and are implicated in autoimmune pathogenesis. The degree to which regulatory variants affect Natural Killer cell gene expression is poorly understood. Here we perform expression quantitative trait locus mapping of negatively selected Natural Killer cells from a population of healthy Europeans ($n = 245$). We find a significant subset of genes demonstrate expression quantitative trait loci specific to Natural Killer cells and these are highly informative of human disease, in particular autoimmunity. A Natural Killer cell transcriptome-wide association study across five common autoimmune diseases identifies further novel associations at 27 genes. In addition to these *cis* observations, we find novel master-regulatory regions impacting expression of *trans* gene networks at regions including 19q13.4, the Killer cell Immunoglobulin-like Receptor region, *GNLY*, *MC1R* and *UVSSA*. Our findings provide new insights into the unique biology of Natural Killer cells, demonstrating markedly different expression quantitative trait loci from other immune cells, with implications for disease mechanisms.

[1] Department of Paediatrics, University of Oxford, Oxford, UK. [2] MRC-Weatherall Institute of Molecular Medicine, University of Oxford, Oxford, UK. [3] Wellcome Centre for Human Genetics, University of Oxford, Oxford, UK. [4] Department of Oncology, University of Oxford, Oxford, UK. [5] Chinese Academy of Medical Science Oxford Institute, University of Oxford, Oxford, UK. [6] NIHR Oxford Biomedical Research Centre, Oxford University Hospitals NHS Foundation Trust, Oxford, UK. [7] These authors contributed equally: James J. Gilchrist, Seiko Makino, Vivek Naranbhai, Julian C. Knight, Benjamin P. Fairfax. ✉email: james.gilchrist@paediatrics.ox.ac.uk; julian.knight@well.ox.ac.uk; benjamin.fairfax@oncology.ox.ac.uk

atural Killer (NK) cells are large granular lymphocytes, comprising 5–15% of peripheral blood lymphocytes, and are key innate effector cells[1]. Functions include cytotoxicity towards virally infected and malignant cells, and production and secretion of cytokines including IFN$\gamma$, defining NK cells as prototypical Group 1 innate lymphoid cells[2]. NK cells express germline-encoded receptors, the best-characterised being Killer cell Immunoglobulin-like Receptors (KIRs), which, dependent on associated intracellular signalling domains, can be activating or inhibitory. Viral infection and dysplasia elicit concomitant induction of stress antigens, triggering activating KIRs, and MHC Class I molecule downregulation, removing the ligands for inhibitory KIRs. Immunodeficiencies distinguished by NK cell deficiency[3–6] or dysfunction[7] are illustrative of the role of NK cells in human health, and are characterised by susceptibility to viral infection, in particular herpesviruses, and early-onset malignancy.

Genome-wide association studies have provided unparalleled insights into the genetic determinants of human health and disease[8]. The majority of associated alleles are non-coding and thought to exert phenotype via the regulation of gene expression. Understanding which cell types and conditions such variants demonstrate activity in is vital to determining pathogenic mechanisms. Expression quantitative trait loci (eQTL) analysis of immune cells isolated from blood has demonstrated high degrees of cellular specificity in regulatory variant function. Whilst analyses of NK cell eQTL have been performed, sample sizes have been small and further data are required[9], especially for the identification of *trans*-regulatory variants which have smaller effect sizes. We describe an eQTL study performed across 245 healthy individuals of Northern European ancestry. We demonstrate that NK cells display many hundreds of eQTL that are not observed in other cell types, and find evidence of NK cell-specific master-regulatory regions. These eQTL colocalise with disease-associated loci, providing novel insights into the role of NK cells in human health and disease.

## Results

***cis*-eQTL mapping**. NK cells were negatively selected from PBMCs from 245 healthy European adults, and gene expression quantified using Illumina gene expression arrays, giving a readout of 18,078 genes post QC. Whole-genome imputation of genome-wide genotyping data yielded high-quality genotypes at 6,012,996 autosomal loci. We defined *cis*-acting variants within 1Mb of the TSS, mapping *cis* eQTL under an additive linear model in QTLtools[10]. To alleviate the effects of confounding variation we included PCs of expression data as covariates in the model, finding 32 PCs maximised *cis* eQTL discovery (Supplementary Fig. 1). We identified *cis* eQTL at 3951 autosomal genes of 17,342 tested (22.7%) at FDR < 0.05 (Fig. 1a, Supplementary Data 1). By conditioning on the peak eSNP at each significant *cis* eQTL we further identified an additional 594 independent *cis* eQTLs at 528 genes; 528 genes have 2 independent eSNPs, 63 genes have 3, and 3 genes have 4 (Fig. 1a, Supplementary Data 2).

We used a Bayesian approach, implemented in moloc[11], to compare the evidence for shared or independent effects at NK eQTL (this dataset, n=245) with CD4$^+$ T cells[12] ($n = 282$), CD8$^+$ T cells[12] ($n = 271$), monocytes[13] ($n = 414$) and neutrophils[14] ($n = 101$). Whilst we find evidence (PP > 0.8) supporting sharing of a single regulatory effect at 2161/3951 (54.7%) NK eQTL with at least one other cell type, the data best supports (PP > 0.8 for a model of association being an eQTL in NK cells alone or independent to eQTLs in other cell types) observed NK eQTL being unique to NK cells at 588/3,951 (14.9%) genes (Fig. 1b, Supplementary Data 3). Amongst these NK-cell-specific eQTL,

270/588 (49.1%) regulate expression of genes for which there are other independent eQTL in at least one other immune cell. Thus, whilst many NK cell-specific eQTL may be reflective of cell-restricted expression, half represent differential regulation of widely expressed genes. In keeping with this observation, there is no significant difference ($p = 0.152$) in median expression of genes with shared or NK cell-specific eQTL (Supplementary Fig. 2).

***cis* eQTL function**. Gene expression is controlled by *cis*-acting regulatory elements and genetic variation within these can lead to regulatory divergence. Correspondingly, eQTL are significantly enriched within regions of active chromatin, promoters, enhancers and TSS[15]. To assess the distribution of *cis* eQTLs in NK cells we plotted densities of observed eQTL around functional chromatin states and transcription factor binding sites, calculating relative enrichment of NK eQTL around each feature (Supplementary Fig. 3, Supplementary Data 4). NK *cis* eQTLs demonstrate clustering around TSS ($p < 2 \times 10^{-8}$), transcribed regions ($p < 2 \times 10^{-8}$), enhancers ($p = 0.0015$) and weak enhancers ($p = 0.0087$), and are underrepresented in regions of the genome with repressed activity ($p < 2 \times 10^{-8}$). Cell-type specific eQTL and conditional eQTL have been demonstrated to show greater overlap with distal regulatory elements, as compared to primary, shared eQTL[16]. In keeping with this, conditional eQTL (secondary, tertiary and quaternary eQTL) in NK cells are enriched for enhancers ($p = 0.0018$), but show no enrichment around promoters (Supplementary Fig. 3). Similarly, while NK cell-specific eQTL demonstrate clustering at promoter sites ($p < 2.2 \times 10^{-7}$) and no significant clustering around enhancers, they demonstrate enrichment for weak enhancers ($p = 0.00053$), significantly more so than NK eQTL shared across immune cells, ($p = 0.0019$, fold change = 3.93).

We similarly observed significant enrichment of overlap between NK cell eQTL and transcription factor binding sites, demonstrating enrichment at 44/52 (84.6%) of transcription factors tested (Supplementary Fig. 3, Supplementary Data 4). The transcription factor binding site enrichment observed across NK cell eQTL encompasses a broad range of transcription factors, many of which are active in immune cells (e.g. NFKB, IRF7, PAX5). In particular, ETS1 has been previously demonstrated to direct NK cell differentiation[17]. We observed no significant enrichment between conditional eQTL in NK cells and transcription factor binding sites. Among NK cell-specific eQTL, we identified significant enrichment of overlap at 14/52 (26.9%) of transcription factor binding sites, including EGR1, POU2F2 and ZEB1 which are predicted to act as transcription factor repressors active in determining ILC subset plasticity[18].

To better understand the biological pathways subject to regulatory variation specific to NK cells, we performed enrichment analysis of genes with an NK cell-specific eQTL (against a background of all tested genes) in XGR[19]. In that analysis, NK cell-specific eQTL are enriched for genes within 26 biological pathways annotated by Gene Ontology Biological Processes (GOBP, Supplementary Data 5, Supplementary Fig. 4). Enriched biological processes highlight the established biology of NK cells, including pathways mediating cell death (GO:0006915, apoptotic process; GO:1902042, negative regulation of extrinsic apoptotic signalling pathway via death domain receptors; GO:0097190, apoptotic signalling pathway; GO:0042981, regulation of apoptotic process; GO:0006919, activation of cysteine-type endopeptidase activity involved in apoptotic process), anti-viral host defence (GO:0009615, response to virus; GO:0051607, defense response to virus), and innate immune signalling (GO:0032496, response to lipopolysaccharide; GO:0033209, tumour necrosis factor-mediated signalling pathway).

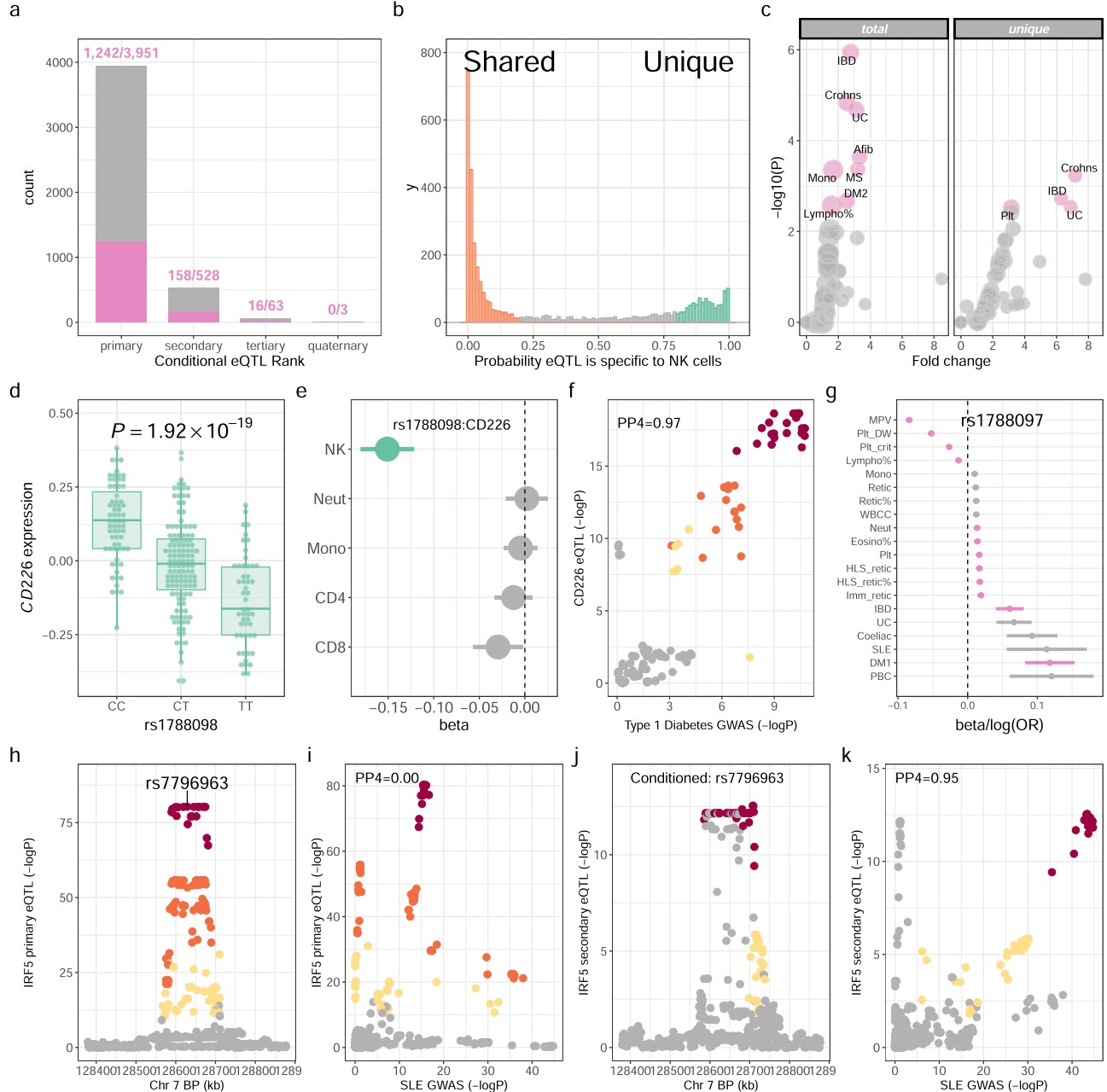

**Fig. 1 *cis* eQTL in NK cells. a** Total significant (*FDR* < 0.05) primary and conditional *cis* eQTL in NK cells from 245 individuals. Numbers of eQTL with evidence of colocalisation with at least one GWAS trait (RTC > 0.9) are highlighted (pink). **b** Frequencies of *cis* eQTL (*FDR* < 0.05) specific to NK cells or shared with other immune cells; monocytes, neutrophils, CD4+ and CD8+ T cells. **c** Enrichment of shared causal loci between NK cell eQTL (total, left panel; NK cell-specific, right panel) and GWAS traits (*n* = 100). Traits are compared with the enrichment observed for height as background. Significant traits (FDR < 0.05) are highlighted (pink). Point size is proportional to a trait's number of GWAS-significant loci. **d** Effect of rs1788098 genotype on *CD226* expression in NK cells (*n* = 245 individuals). Box and whisker plot; boxes depict the upper and lower quartiles of the data, and whiskers depict the range of the data excluding outliers (outliers are defined as data-points > 1.5 × the inter-quartile range from the upper or lower quartiles). **e** The effect of rs1788098 genotype on *CD226* expression is specific to NK cells. Significant eQTL effects are highlighted (green). **f** The *CD226* eQTL in NK cells colocalises with a risk locus for type-1 diabetes. SNPs are coloured according to strength of LD (CEU population) to the peak eSNP (rs1788098); brown $r^2$ > 0.8, orange 0.5 < $r^2$ ≤ 0.8, yellow 0.2 < $r^2$ ≤ 0.5, grey $r^2$ ≤ 0.2. **g** Association of rs1788097 (exact proxy for rs1788098 in European populations, $r^2$ = 1) with autoimmune diseases and haematological indices. GWAS-significant ($p$ < 5 × 10$^{-8}$) associations are highlighted (pink). **h** Regional association plot of the primary *IRF5* eQTL in NK cells. **i** The primary *IRF5* eQTL does not colocalise with a GWAS risk locus in the *IRF5* region for systemic lupus erythematosus. **j** Conditioning on the peak *IRF5* eSNP reveals an independent, secondary eQTL for *IRF5* in NK cells. **k** The secondary *IRF5* eQTL colocalises with the *IRF5* region for systemic lupus erythematosus. Genotype to phenotype correlations were calculated with linear regression. *P* values are two-sided.

***cis* eQTL disease associations.** To investigate the disease informativeness of NK cells we adopted two complementary approaches. We firstly used Regulatory Trait Concordance (RTC)[20] to determine the proportion of NK cell eQTL predicted to share a causal variant with NHGRI-EBI GWAS Catalog studies (*n* = 5823). Among primary *cis* NK cell eQTL, 1242/3951 (31.4%) eQTL are predicted to share the same functional variant with at least one GWAS trait (Fig. 1a, Supplementary Data 6). Among

eQTL predicted to be unique to NK cells, 161/588 (27.4%) loci are predicted to colocalise with at least one GWAS-significant locus. Conditional eQTL are similarly informative of human disease and GWAS traits, with 174/594 (29.3%) conditional eQTL in NK cells colocalising with GWAS trait-associated loci.

Next, to better understand the impact that regulatory variation modifying gene expression in NK cells has on human health and disease, we used evidence of colocalisation between NK cell eQTL and GWAS traits, derived with coloc[21], to estimate the degree of enrichment of NK eQTL colocalisation with GWAS traits as compared to that expected by chance. Height GWAS loci are enriched for regulatory variation operating in connective tissue and mesenchymal stem cells but not immune cells[22], and any shared causal variants with NK cell eQTL are likely to be representative of the proportion one would expect by chance (i.e. shared causal loci are likely to represent effects mediated by other cell types)[23]. We therefore used a well-powered GWAS of height in European populations (UK Biobank, URL: http://www.nealelab.is/uk-biobank/) as our background against which to compare enrichment in other traits. We performed enrichment analysis across 100 traits (Supplementary Data 2); 40 UK Biobank continuous traits (anthropometrics and haematological indices) and 60 NHGRI-EBI GWAS Catalog traits (well-powered, case-control studies in populations of European ancestry). In that analysis, we identify 8 GWAS traits with significant enrichment (FDR < 0.05) of colocalisation with NK cell eQTL, and 4 traits with significant enrichment of colocalisation with NK cell-specific eQTL (Fig. 1c, Supplementary Data 7). Among both NK cell-specific and shared eQTL, we observe marked enrichment of shared causal loci with autoimmune diseases (inflammatory bowel disease, multiple sclerosis, type-1 diabetes) and a range of haematological indices including platelet, monocyte and lympho-cyte count (Fig. 1c). To investigate whether the choice of height as background for enrichment could bias our results, we recapitulated our analysis using whole-body impedance (UK Biobank phenotype code: 23106_irnt) as an alternative background. In that analysis we find no evidence that choice of background significantly affects our results, with our enrichment estimates derived using height as background being well-correlated ($r = 0.575$, $p = 5.51 \times 10^{-6}$) with those using whole-body impedance (Supplementary Fig. 5).

Genes with NK cell-specific eQTL include *CD226* (Fig. 1d, e), which encodes CD226 (also known as DNAX accessory molecule-1), a surface-expressed immunoglobulin superfamily glycoprotein, widely expressed on NK cells[24]. CD226 engages CD155 on antigen-presenting cells, an interaction increasing cytotoxicity and IFNγ production, that is competitively inhibited by NK-expressed TIGIT and CD96[25]. The NK cell-specific eQTL for *CD226* (peak eSNP: rs1788098, $p = 1.92 \times 10^{-19}$) colocalises with genetic loci for a range of haematological indices and autoimmune diseases; including IBD and T1DM (Fig. 1f, g). The direction of effect of *CD226* expression on autoimmune disease risk is of reduced expression increasing disease risk, suggesting less robust NK cell responses (potentially directed at viral pathogens or autoreactive T cells) increases the risk of autoimmunity. Indeed, NK cells have been shown to restrain the activity of CD4+ T cells in a CD226-dependent manner in the context of multiple sclerosis[26]. Other NK cell-specific eQTL informative for human disease include; the caspase protease *CASP8* (peak eSNP: rs3769821, $p = 1.24 \times 10^{-8}$) which colocalises with risk loci for multiple malignancies (breast cancer, non-small cell lung cancer, melanoma, oesophageal squamous cell carcinoma), and the fucosyltransferase *FUT11* (peak eSNP: rs11000765, $p = 1.91 \times 10^{-18}$) which colocalises with asthma and lung function.

As expected, NK eQTL shared with other immune cells are also highly informative with respect to human health and disease. For instance, an *ERAP2* eQTL (peak eSNP: rs1363974, $p = 1.33 \times 10^{-76}$), predicted to be shared across all immune cells tested (NK cells, CD4+ and CD8+ T cells, monocytes and neutrophils), is a genetic determinant of neutrophil and lymphocyte percentage (Supplementary Fig. 6). As described above, conditional eQTL are also informative with regard to human disease risk (Supplementary Data 2). A secondary eQTL for *IRF5*, encoding the transcription factor IRF5, a positive regulator of type-1 interferon production, provides an example of this (Fig. 1h–k). Here the primary eQTL at *IRF5* (peak eSNP: rs7796963, $p = 2.37 \times 10^{-65}$) has no evidence of colocalisation with human disease traits, however the secondary eQTL (peak eSNP: rs17424921, $p = 5.39 \times 10^{-9}$) shows strong evidence of a shared causal variant with systemic lupus erythematosus.

**trans eQTL**. Detecting *trans* eQTL within purified cell types can elucidate cell-specific regulatory networks[13]. We defined *trans*-acting regulatory variation as variants > 5Mb from the TSS, and tested for associations at 18,078 genes under an additive linear model in QTLtools[10], alleviating confounding variation through the inclusion of expression data PCs as covariates in the model (12 PCs maximising *trans* discovery—Supplementary Fig. 1). We identified 84 independent loci (totalling 2,266 SNPs) demonstrating *trans* effects (FDR < 0.05) to 64 genes (Fig. 2a, Supplementary Data 8). Regulatory variation acting in *trans* is enriched for *cis* eSNPs[27,28], suggesting that a proportion of *trans*-acting variation operates via *cis* effects on upstream regulatory genes. Consistent with this, of the 64 genes with *trans* eQTL, 25 have evidence of *cis* mediation (colocalisation probability of *cis* and *trans* effect > 0.8) from 16 genes (Supplementary Data 8).

Among *trans*-acting loci, 7 *trans* eSNPs affect more than one gene distally, and show master-regulatory properties. For instance, a *trans*-regulatory network mediated by a NK-cell-specific *cis* eQTL for *GNLY* (peak eSNP: rs1866140, $p = 8.86 \times 10^{-23}$), determines expression of 5 genes in *trans*; *ARHGAP30*, *EDNRB*, *KIAA1586*, *N4BP2*, *NOP56* (Fig. 2b–d). *GNLY* encodes granulysin, a secreted cytolytic molecule with broad antimicrobial activity and cytotoxicity towards infected and malignant cells[29]. In addition to cytolytic activity, granulysin also induces immune cell chemotaxis and pro-inflammatory cytokine production in monocytes. The expression of a secreted molecule as a *cis* mediator of a *trans*-regulatory network within a cell type is highly analogous to the previously-described *trans* eQTL network mediated by *LYZ*, encoding the bactericidal gene lysozyme[30–32], and supports a model in which the expression of key secreted genes is associated with gene expression networks in *trans*.

A second example of a *trans*-regulatory network, affecting *DRD3*, *FAM169B*, *JADE1*, *PDHA2* and *SNORD85* expression, is mediated in *cis* by an eSNP affecting *MC1R* expression (Fig. 2e–g, peak eSNP: rs11642267, $p = 5.80 \times 10^{-35}$). The *MC1R cis* eQTL is shared with CD4+ and CD8+ T cells (Supplementary Data 5), and the lead SNP in the non-imputed genotyping was rs2228479, encoding a non-synonymous polymorphism (V92M) within *MC1R* and in perfect LD ($R^2 = 1$) with rs11642267. Given this, we tested for allele-specific expression of this allele across five individuals heterozygous for rs2228479, using the C-BASE assay[33,34], observing for all individuals significantly increased expression of the minor allele ($p = 0.01$–$p = 2.3 \times 10^{-7}$, combined $p < 2.2 \times 10^{-16}$, $\chi^2$ test). Notably no ASE was observed in monocytes at this SNP (Supplementary Fig. 7). *MC1R* encodes the melanocortin-1 receptor (MC1R), a high-affinity receptor for α-melanocyte-stimulating hormone (αMSH), and most robustly expressed in melanocytes, with associations with tanning[35], freckling, hair and eye colour[36], melanoma[37] and non-melanoma

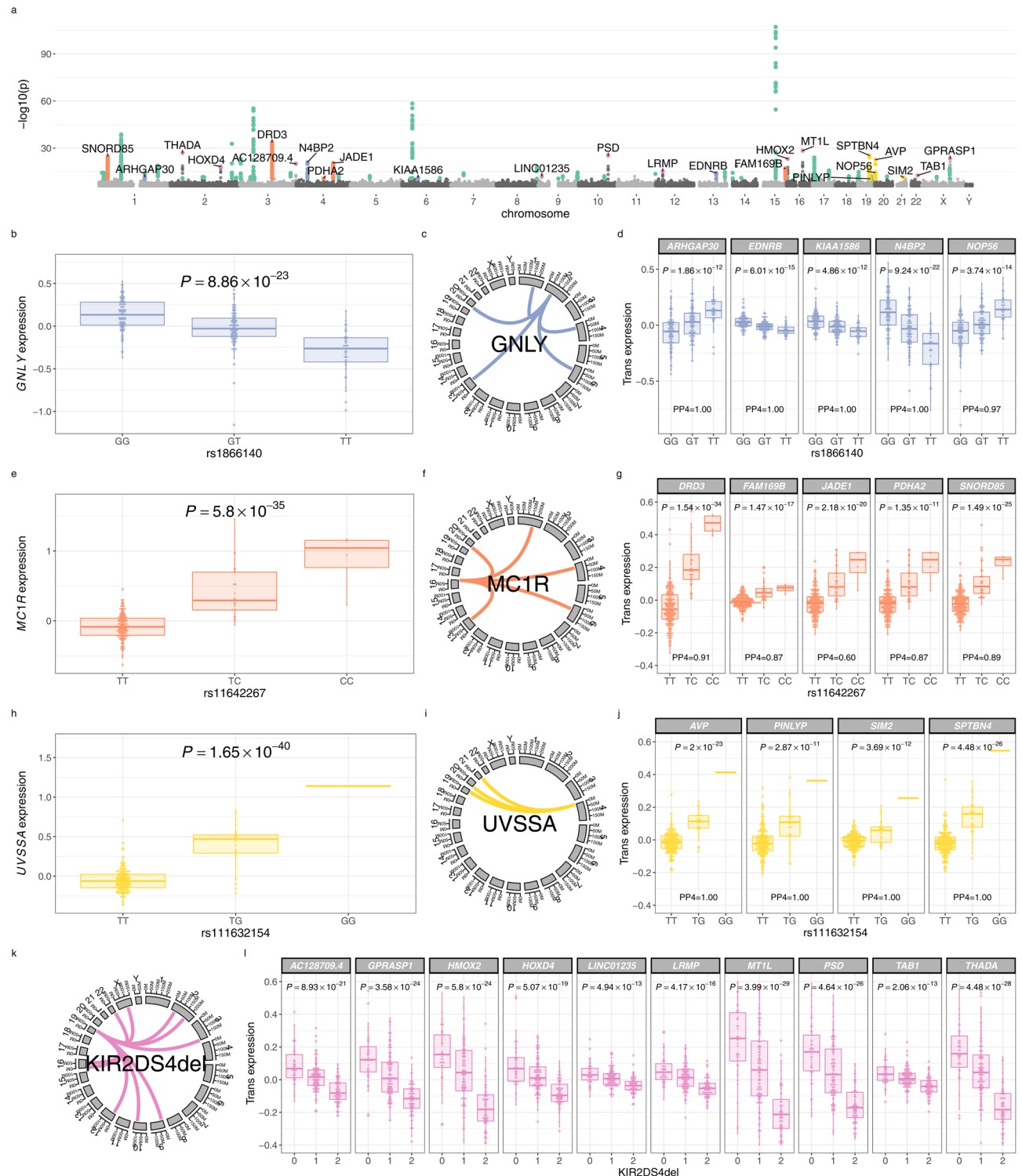

skin cancer[38]. As well as its effects on pigmentation, MC1R/αMSH signalling has well-established immunomodulatory functions, and MCR1 is surface expressed on B cells, T cells and NK cells[39]. Whilst the effect of MC1R/αMSH signalling in NK cells is undefined, NK cell dysfunction, as part of broader immunodeficiency, is well-characterised as part of syndromic disorders of hypopigmentation, e.g. Chediak-Higashi, Hermansky-Pudlak and Griscelli syndromes[40]. All of these syndromes are characterised by impaired secretory lysosome function, impairing melanin secretion from melanocytes as well as cytotoxic granule secretion from NK cells and cytotoxic T cells. Our description here of an NK cell

trans-regulatory network mediated by MC1R expression suggests that the NK cell immune defect seen in syndromic disorders of hypopigmentation may not be confined to defective cytotoxic degranulation.

We also observe a *trans*-regulatory network, affecting *AVP*, *SPTBN4*, *PINLYP* and *SIM2* expression, which is mediated in *cis* by an eSNP affecting *UVSSA* expression (Fig. 2h–j, peak eSNP: rs111632154, $p = 1.65 \times 10^{-40}$). *UVSSA* encodes UV-sensitivity scaffold protein A, a cofactor for RNA polymerase II, which facilitates the release of transcriptional machinery from damaged, transcriptionally active DNA, allowing transcription-coupled

**Fig. 2 *trans* eQTL in NK cells. a** Manhattan plot depicting significant *trans* eQTL in NK cells. Physical position (*x* axis) represents location of the target gene. Coloured points represent significant (FDR < 0.05) *trans* eQTL; 2266 *trans* SNP-gene associations, affecting expression of 64 genes. Regulatory networks are highlighted as follows; *GNLY* (blue), *MC1R* (orange), *UVSSA* (yellow), KIR (pink), all others (green). **b** Effect of rs1866140 genotype on *GNLY* expression in NK cells (*n* = 245 individuals). **c** A *trans*-regulatory network of 5 genes, mediated in *cis* by *GNLY* expression. **d** Effect of rs1866140 genotype on *GNLY* regulatory network genes in *trans* (*n* = 245 individuals). PP4, posterior probability of a shared causal variant with *GNLY cis* eQTL. **e** Effect of rs117406136 genotype on *MC1R* expression in NK cells (*n* = 245 individuals). **f** A *trans*-regulatory network of 4 genes, mediated in *cis* by *MC1R* expression. **g** Effect of rs117406136 genotype on *MC1R* regulatory network genes in *trans* (*n* = 245 individuals). PP4, posterior probability of a shared causal variant with *MC1Rcis* eQTL. **h** Effect of rs111632154 genotype on *UVSSA* expression in NK cells (*n* = 245 individuals). **i** A *trans*-regulatory network of 4 genes, mediated in *cis* by *UVSSA* expression. **j** Effect of rs111632154 genotype on *UVSSA* regulatory network genes in *trans* (*n* = 245 individuals). PP4, posterior probability of a shared causal variant with *UVSSA cis* eQTL. **k** A *trans*-regulatory network of 10 genes, mediated mediated by KIR2DS4del. **l** Effect of KIR2DS4del genotype on KIR regulatory network genes in *trans* (*n* = 245 individuals). Box and whisker plots; boxes depict the upper and lower quartiles of the data, and whiskers depict the range of the data excluding outliers (outliers are defined as data-points > 1.5 × the inter-quartile range from the upper or lower quartiles). Genotype to phenotype correlations were calculated with linear regression. *P* values are two-sided.

---

nucleotide excision repair[41–43]. Mutations in *UVSSA* underlie UV-sensitive syndrome, an autosomal recessive disorder characterised by cutaneous photosensitivity without predisposition to malignancy[44]. The *UVSSA cis* eQTL has strong evidence of colocalisation with bone mineral density GWAS loci (*PP4* = 1). In keeping with this, other defects of transcription-coupled nucleotide excision repair, e.g. Cockayne syndrome, are characterised by features of accelerated aging, including osteoporosis[45].

We also identify a *trans*-regulatory network for which the master regulator maps to the leucocyte receptor complex (LRC, 19q13.4), which encodes KIR genes. Expression of 3 genes (FDR < 0.05) in *trans* to the LRC are regulated by a KIR region SNP: rs640345 (Table S8). The LRC is highly polymorphic in terms of KIR gene content and allelic variation. To better define the *cis* mediator of this regulatory network, we imputed KIR copy number in the study samples using KIR*IMP[46] and remapped *trans* eQTL in NK cells using KIR copy number imputations. KIR gene/pseudogene copy numbers were well-imputed (Table S9), with estimated imputation accuracies ranging from 76.2% (KIR2DS2) to 96.9% (KIR2DS1) and high levels of concordance with gene presence/absence as determined by PCR (80.7–100%). In that analysis, we find the strongest evidence for *trans*-regulatory activity at KIR2DS4del: a 22 base-pair deletion in *KIR2DS4* which introduces a premature stop codon[47]. KIR haplotypes are broadly grouped into A and B haplotypes, and for A haplotypes KIR2DS4 is the only activating receptor. KIR2DS4 ligands have been challenging to identify but include HLA-C*05:01-bound peptides derived from epitopes common to a number of bacterial pathogens[48]. KIR2DS4del has significant regulatory effects on a *trans* network of 10 genes (Supplementary Data 10, Fig. 2k, l). KIR2DS4del is the lead regulatory variant for all 10 genes within the KIR *trans*-regulatory network (Supplementary Data 10), with no significant independent associations remaining after conditioning on KIR2DS4del copy number (Supplementary Data 10). In addition, KIR imputation revealed a second KIR *trans*-regulatory network independent of KIR2DS4-del (Supplementary Data 10). That network comprises 3 genes (*MYRF*, *EGLN3*, *SYNGR3*) regulated by KIR3DP1 copy number.

We further sought to replicate evidence for our *trans* associations in NK cells in the large-scale meta-analysis of whole blood eQTL data compiled by the eQTLGEN consortium[49]. NK cells represent a small proportion of the circulating leucocytes in whole blood. Given that *trans* eQTL hubs are frequently thought to be cell-type specific[49], our expectation was that our power to replicate *trans* associations in whole blood eQTL would be limited. Despite this, we were able to replicate a small number of NK cell *trans* associations observed in our study in the eQTLGEN consortium data (Supplementary Data 8). We find evidence of replication (eQTLGEN experiment-wide FDR < 0.05) for the

effect of rs3811444 on *JAM3* expression and for rs10876864 on *KCTD11* expression in whole blood (Supplementary Data 8). In both cases the replicated *trans* association forms part of a larger *trans*-regulated gene network in whole blood, with rs3811444 regulating the expression of 97 genes in *trans* and rs10876864 the expression of 47. Moreover, while there is no direct evidence for replication of the *trans* network mediated by *MC1R* expression, rs9939914 (a significant *trans* eSNP for both *DRD3* and *SNORD85* in our data) affects the expression of a network of four genes in *trans* in whole blood; *GPR25*, *PHF17*, *LDHD*, *BBS10*. These observations are in keeping with a model in which, while some *trans*-regulatory eSNPs are likely to be shared across cell type and context, the gene network which they regulate will vary according to cell type. In keeping with this, in the case of the *trans*-regulatory hub including rs10876864, for which we see evidence of replication of its effect on *KCTD11* expression in whole blood, we have previously reported *trans* effects at this locus on *LAP3P2*, *IP6K2* and *HELZ2* in B cells and *LAP3P2* in monocytes[31].

**TWAS**. A striking feature of our eQTL analysis was the degree of enrichment for autoimmune disease that we observed among *cis* eQTL in NK cells. To further investigate this we utilised NK cell eQTL data and GWAS summary statistics for autoimmune disease to identify novel autoimmune risk genes within five autoimmune TWAS; ulcerative colitis[50], rheumatoid arthritis[51], systemic lupus erythematosus[52], primary biliary cirrhosis[53] and Crohn's disease[50]. We identified 98 TWAS-significant, independent gene-trait associations (FDR < 0.05) for autoimmune diseases (Fig. 3, Supplementary Data 11–15); ulcerative colitis 21/2998 genes tested, rheumatoid arthritis 5/3015 genes tested, systemic lupus erythematosus 18/3045 genes tested, primary biliary cirrhosis 18/3035 genes tested, Crohn's disease 31/3077 genes tested. We then sought to define which of these gene-trait pairs are explained by an NK cell-specific *cis* eQTL, that is instances in which the most likely model includes an eQTL signal unique to NK cells (see "Methods") and the eQTL in NK cells colocalises with the GWAS signal (PP > 0.8). In that analysis, 6 gene-traits pairs are driven by gene expression in NK cells specifically; *CD226* expression modifies systemic lupus erythematosus and primary biliary cirrhosis risk, *TMEM163* expression modifies primary biliary cirrhosis risk, *CALHM6* expression modifies ulcerative colitis risk, and both *ZMIZ1* and *OSM* modify Crohn's disease risk. We considered a gene-trait association to be novel if there was no GWAS-significant locus within 1Mb of the gene in the trait's GWAS used as input for the TWAS, and if there was no colocalisation evidence (L2G score ≥ 0.3) supporting gene-trait association in Open Targets Genetics[54] (URL: https://genetics.opentargets.org, accessed 28/01/2021). Of the 98 TWAS-significant trait-associated genes, 27 define novel gene-trait associations (Table 1).

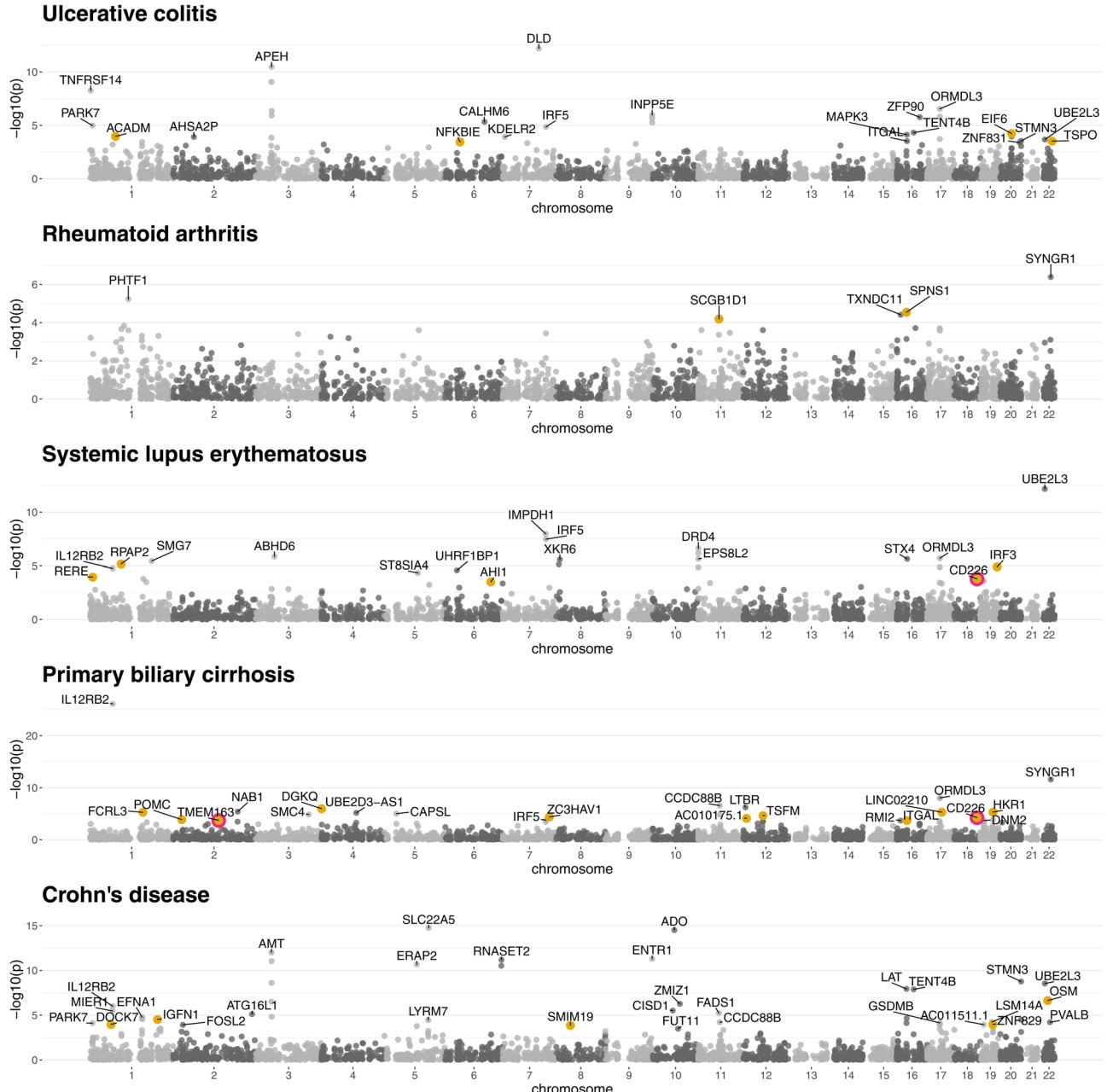

**Fig. 3 NK cell TWAS in autoimmune diseases.** Manhattan plot of autoimmune disease TWAS. All significant trait-associated genes ($n = 98$) are labelled. Novel gene-trait associations (no GWAS-significant locus within 1Mb and no evidence supporting gene-trait association in Open Targets Genetics) are highlighted (yellow, $n = 27$). Novel gene-trait associations which colocalise (posterior probability colocalisation > 0.8) with an NK cell-specific eQTL are circled (pink, $n = 3$).

We were particularly interested in identifying novel gene-trait associations for which the evidence supports a shared causal variant between the trait GWAS and an eQTL specific to NK cells (defining colocalisation and NK cell specificity as above). We were able to identify three instances of this. Complementary to our identification of NK cell expression *CD226* as a risk factor for type-1 diabetes and multiple sclerosis (Fig. 1), we identify NK cell expression of *CD226* as a determinant of both systemic lupus erythematosus and primary biliary cirrhosis. We also identified NK cell expression of *TMEM163* as a determinant of primary biliary cirrhosis. *TMEM163* encodes a transmembrane zinc transporter[55], which is hypothesised to mediate zinc accumulation into lysosomes[56]. Notably, zinc deficiency has been reported to be associated with reduced NK cell cytolytic activity[57]. Genetic

variation affecting *TMEM163* expression has not previously been implicated in the pathogenesis of autoimmune disease. However, in an interesting parallel to our description of a *trans*-regulatory network in NK cells mediated by *MC1R* expression, eQTL for *TMEM163* in whole blood[49] colocalise (PPH4 = 0.96, Open Targets Genetics) with genetic determinants of hair colour (UK Biobank, URL: http://www.nealelab.is/uk-biobank/). *TMEM163* is not regulated in *trans* by the *MC1R* locus in NK cells, however, these data suggest a model in which the shared biological determinants of pigmentation and inflammation operate in NK cells and modify risk of immune-mediated disease. Immunodeficiencies with combined NK cell dysfunction and disorders of pigmentation, including Chediak–Higashi, Hermansky–Pudlak and Griscelli syndromes[40], are characterised by impairment of

**Table 1 Novel gene-trait associations identified by TWAS.**

| GWAS trait | Peak GWAS SNP | Gene | NK cell cis eSNP | TWAS Z score | TWAS P value |
|---|---|---|---|---|---|
| Ulcerative colitis | rs6060341 | EIF6 | rs6120889 | 4.019 | $5.85 \times 10^{-5}$ |
| | rs4949874 | ACADM | rs2211080 | −3.874 | 0.0001 |
| | rs6971 | TSPO | rs138931 | −3.619 | 0.0003 |
| | rs6458351 | NFKBIE | rs28385699 | 3.568 | 0.0004 |
| Rheumatoid arthritis | rs7500321 | SPNS1 | rs7140 | 4.185 | $2.85 \times 10^{-5}$ |
| | rs968567 | SCGB1D1 | rs174627[a] | 3.996 | $6.44 \times 10^{-5}$ |
| Systemic lupus erythematosus | rs12758175 | RPAP2 | rs7522081 | 4.499 | $6.83 \times 10^{-6}$ |
| | rs7258381 | IRF3 | rs12104272 | −4.370 | $1.24 \times 10^{-5}$ |
| | rs172531 | RERE | rs159963 | 3.867 | 0.0001 |
| | **rs727088** | **CD226** | **rs1788098** | **−3.77** | **0.0002** |
| Primary biliary cirrhosis | rs3796621 | DGKQ | rs3733349 | 4.874 | $1.09 \times 10^{-6}$ |
| | rs12462708 | HKR1 | rs7257354 | 4.569 | $4.90 \times 10^{-6}$ |
| | rs1876829 | LINC02210 | rs77459448 | −4.564 | $5.03 \times 10^{-6}$ |
| | rs3761959 | FCRL3 | rs2210913 | −4.551 | $5.34 \times 10^{-6}$ |
| | rs11172113 | TSFM | rs870392[a] | 4.227 | $2.37 \times 10^{-5}$ |
| | rs6945033 | ZC3HAV1 | rs7800079 | −4.097 | $4.19 \times 10^{-5}$ |
| | **rs1788103** | **CD226** | **rs1788098** | **−4.026** | **$5.66 \times 10^{-5}$** |
| | rs33873 | AC010175.1 | rs35596029 | −3.951 | $7.77 \times 10^{-5}$ |
| | rs7575363 | POMC | rs28445639 | −3.819 | 0.0001 |
| | **rs1467194** | **TMEM163** | **rs6758396** | **3.727** | **0.0002** |
| | rs4465620 | ITGAL | rs12598978 | 3.712 | 0.0002 |
| Crohn's disease | rs2412973 | OSM | rs201712052 | 5.174 | $2.30 \times 10^{-7}$ |
| | rs7522462 | IGFN1 | rs940398[a] | 4.174 | $3.00 \times 10^{-5}$ |
| | rs4409689 | DOCK7 | rs144467554 | −3.869 | 0.0001 |
| | rs1035441 | ZNF829 | rs1148395[a] | 3.871 | 0.0001 |
| | rs2974298 | SMIM19 | rs2974348 | 3.809 | 0.0001 |

Loci where the data support a shared causal variant between the trait GWAS and an eQTL specific to NK cells are highlighted (bold).
[a]Peak eSNP not significant in primary eQTL analysis.

secretory lysosome function. Given the described role of *TMEM163* in lysosome composition, it is tempting to speculate that *TMEM163* may modify both pigmentation and risk of immune-mediated disease through effects on melanin secretion and NK cell cytotoxic degranulation.

**Protein phenotypes of NK cell eQTL.** Mapping of regulatory genetic variation has been largely undertaken in healthy individuals. Moreover, our eQTL mapping data defines regulatory genetic variation as a determinant of RNA expression, but does not provide direct evidence for genetic control of NK cell phenotypes at the protein level. To better understand the relevance of eQTL mapping data to protein level expression of key NK cells markers, and to demonstrate the relevance of eQTL mapping data to a disease context, we correlated genotype at NK cell eSNPs with expression of key surface markers on NK cells from patients with metastatic melanoma[58]. Using multiparametric flow cytometry, we sought to confirm the effect of identified NK cell eSNPs on CD226 and CD57 (*B3GAT1*-encoded) protein expression. We further correlated KIR2D antigen expression with KIR2DS4del copy number, which we have defined as a determinant of a large *trans*-regulatory network.

T cell and NK cell-expressed CD226 is a determinant of anti-tumour responses[59], including in the context of metastatic melanoma[60]. In a separate cohort of patients with metastatic melanoma, we confirm that that the peak eSNP for *CD226* is a determinant of CD226/DNAM1 protein expression on NK cells ($p = 0.03$, Fig. 4). It has previously been shown that the balance between CD226 and TIGIT (a competitive inhibitor of CD226) expression on $CD4^+$ Tregs is a correlate of clinical response to immune checkpoint blockade in metastatic melanoma[61]. Our identification of an NK cell-specific eQTL, which modulates CD226 protein expression on NK cells, provides the opportunity for future studies to leverage genetic variation to better understand the cellular context of CD226 expression as a determinant of response to immune checkpoint blockade.

CD57 expression defines a subpopulation of mature, cytotoxic, $CD56^{dim}$ NK cells[62]. Our eQTL mapping analysis identified a *cis* eQTL for *B3GAT1* (which encodes CD57) expression in NK cells (peak eSNP: rs77478906, $p = 1.90 \times 10^{-11}$). This eQTL is predicted to be specific to NK cells, and colocalises with a GWAS[63] risk locus for prostate cancer ($PP4 = 0.949$), with decreased NK cell *B3GAT1* expression being associated with increased prostate cancer risk. We confirm the effect of rs77478906 on CD57 NK cell protein expression in patients with metastatic melanoma ($p = 3.85 \times 10^{-5}$, Fig. 4), demonstrating that the prostate cancer risk allele, rs77478906:C, is associated with a decreased proportion of $CD57^+$ NK cells. The proportion of $CD57^+$ NK cells have been observed to increase with age[64] and CMV infection[65], but it remains unclear how this interacts with malignancy risk or prognosis. Our study demonstrates that regulatory genetic variation is also a key determinant of the $CD57^+$ NK cell proportion, and highlights that studies seeking to understand the relationship between $CD57^+$ NK cells, age, CMV and malignancy should also consider rs77478906 genotype.

Finally, having identified KIR2DS4del copy number as a mediator of a *trans*-regulatory network in NK cells, we sought to confirm a role for imputed KIR2DS4del copy number on KIR antigen expression in the same patient cohort. In metastatic melanoma patients, KIR2DS4del copy number is associated with the proportion of $KIR2D^+$ NK cells ($p = 0.046$, Fig. 4), with increased copies of the deletion being associated with decreased KIR2D expression. It has previously been demonstrated that a population of $CD56^{dim}CD57^+CD69^+CCR7^+KIR^+$ NK cells are expanded in tumour infiltrated lymph nodes of patients with melanoma, and that these NK cells are key effectors of anti-tumoral cytotoxicity[66]. Our data suggest that proportions of $CD57^+$ and $KIR^+$ NK cells in the periphery are in part genetically

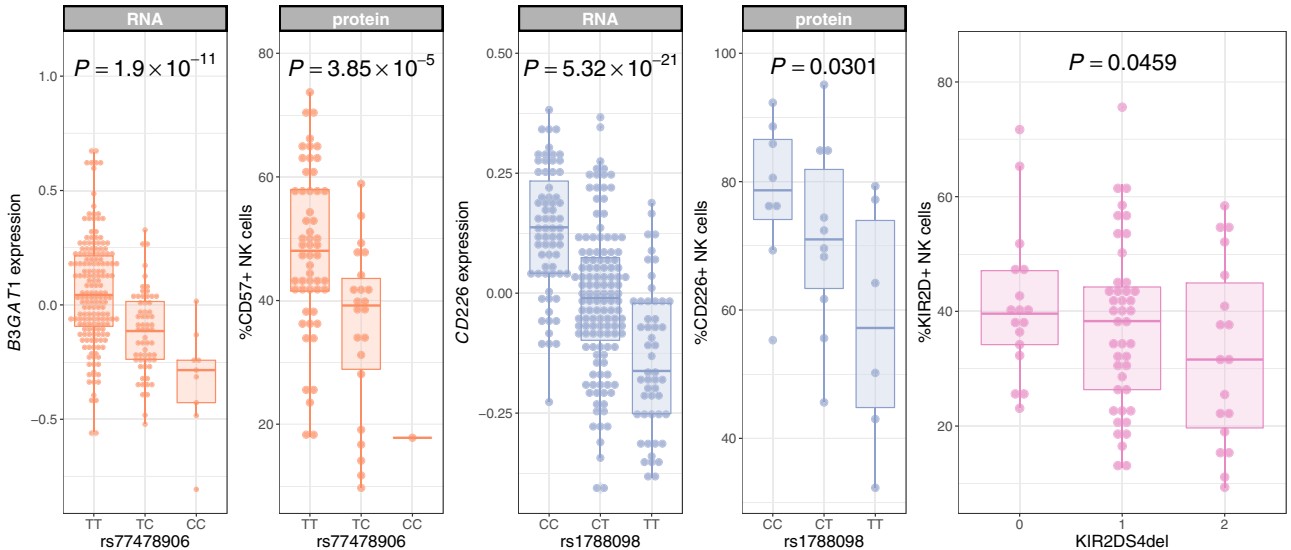

**Fig. 4 NK cell eQTL determine protein expression on NK cells in patients with metastatic melanoma.** Genotype-phenotype correlations are calculated by linear regression. *P* values are calculated with likelihood ratio tests. Expression of CD57 (orange, *n* = 78) and CD226/DNAM1 (blue, *n* = 24) protein on NK cells is determined by the respective RNA eSNPs in patients with metastatic melanoma. KIR2D antigen expression on NK cells in patients with metastatic melanoma (*n* = 78) is modified by KIR2DS4del copy number (pink). Box and whisker plots; boxes depict the upper and lower quartiles of the data, and whiskers depict the range of the data excluding outliers (outliers are defined as data-points > 1.5 × the inter-quartile range from the upper or lower quartiles). Genotype to phenotype correlations were calculated with linear regression. *P* values are two-sided.

determined, and that in the case of KIR positivity, this is associated with large-scale transcriptional changes in *trans*. A complete understanding of NK cell phenotypes as a predictor of cancer predisposition or outcome needs to reflect genetically determined inter-individual differences in NK cell phenotype at baseline.

## Discussion
Knowledge of the genetic determinants of gene expression in immune cells has proved highly informative in advancing our understanding of immune-mediated disease. Here, we have defined the regulatory landscape of gene expression in primary human NK cells. In doing so, we identified thousands of eQTL operating in NK cells, and demonstrated that a significant proportion of this regulatory variation appears specific to these cells.

Regulatory variation active in NK cells is highly enriched for disease-associated genetic variation. By combining our eQTL mapping with colocalisation analysis, incorporating the results of multiple genetic association studies, we elucidate many examples where regulatory variation specific to NK cells are associated with human disease risk. As such, our data directly implicate gene expression in NK cells in a range of human traits and diseases, most strikingly for autoimmune diseases. We expand on these findings, leveraging our NK eQTL data to perform TWAS in five autoimmune diseases, defining novel roles for NK cell expression of *CD226* and *TMEM163* in systemic lupus erythematosus and primary biliary cirrhosis. Furthermore, we illustrate the effects of identified NK cell eSNPs on the expression of key NK cell markers at the protein level in a separate cohort of well-characterised patients with metastatic melanoma, replicating these observations at the protein level and illustrating the potential relevance to disease.

Our findings are in keeping with an emerging role for NK cells in autoimmune disease. This role is supported by observational data across a number of autoimmune diseases, identifying decreased NK cell number in the periphery as a marker of disease risk or activity[67–69], and accumulation of NK cells at sites of autoimmunity[70,71]. The presence of NK cells at sites of

autoimmunity could suggest a model in which NK cells directly contribute to tissue damage through cytolysis and/or cytokine secretion, or alternatively a model in which NK cells have an immunoregulatory role, restraining the activity of autoreactive immune cells. Recent data have highlighted the importance of the latter model, defining immunoregulatory roles for NK cells in controlling the activity of CD4[+] T cells[26,72].

We identified multiple *trans* eQTL operating in NK cells, most notably *trans*-regulatory networks mediated by MC1R, GNLY, UVSSA, and KIR regions. The identification of a large *trans*-regulatory network mediated by a common deletion in an activating KIR receptor (KIR2DS4del) is particularly intriguing and is analogous to previous descriptions of master-regulatory variation to the MHC[31,73]. Variation in KIR type has been implicated in the pathogenesis of infectious and autoimmune disease. It is highly likely that the KIR-mediated *trans*-regulatory network we have identified here will have important roles in human disease risk, and that this role will emerge as the KIR region is better defined, through imputation[46] or sequencing[74] approaches, in GWAS cohorts.

Our study has some limitations. Most notably, the expression data we present here was generated with microarray technology rather than with RNA sequencing. Large-scale meta-analysis of eQTL mapping in whole blood have demonstrated excellent concordance between studies performed with microarray and RNASeq[49]. However, within this dataset we are unable to identify genetic mediators of expression in NK cells for transcripts lacking probe coverage, alternative splicing and exon/transcript usage.

In summary, we have described the *cis*- and *trans*-regulatory landscapes of gene expression in human NK cells. We have integrated these data with epigenetic and GWAS datasets deriving important insights into NK cell biology and its impact on human health and disease.

## Methods

**Study samples and power**. Healthy individuals of European ancestry (*n* = 245) were recruited following written informed consent (Oxfordshire Research Ethics Committee COREC reference 06/Q1605/55). Participants had a median age of 28 years (range 18–66) and 117 of 245 were male. PBMCs, purified by Ficoll-Pacque

density gradient centrifugation, were isolated from whole blood collected into EDTA-containing tubes (Vacutainer system, Becton Dickinson). Cells were washed in Hanks' balanced salt solution (Invitrogen) and enumerated with a haemocytometer.

CD56$^+$CD3$^-$ NK cells were isolated by magnetic-activated cell sorting negative selection (MACS, cat 130-092-657, Miltentyi Biotec). Cells snap-frozen in RLT reagent (Qiagen) prior to RNA extraction. Total RNA was extracted using RNAeasy Mini kits (Qiagen), quantified using a NanoDrop (ThermoFisher), and BioAnalyzer quantification in a subset.

We estimated study power to detect trait-associated genetic variation in *cis* and *trans* across a range of minor allele frequencies and numbers of independent tests (Supplementary Fig. 8). In each case we used powerEQTL[75] to calculate power for $\alpha = 0.05$/number of tests and an effect size and standard deviation of 0.13[76]. For *cis* power calculations we assume a variable number of independent tests, defined as the number of independent SNPs in a *cis* testing window (1–5000) multiplied by the number of genes tested (18,000). For *trans* power calculations we assume $18 \times 10^9$ independent tests, assuming 18,000 genes and $1 \times 10^6$ independent SNPs genome-wide. Assuming an effective sample size of 245, our study has 80% power to detect a *cis* eQTL with a causal eSNP with minor allele frequency > 0.08, assuming 100 independent tests in the *cis* testing window. In *trans*, our study has 80% power to detect a *trans* eQTL with a causal eSNP with minor allele frequency > 0.12.

Patients with metastatic melanoma ($n = 78$) were recruited as described previously[58]. Patients provided blood samples to the Oxford Radcliffe Biobank (Oxford Centre for Histopathology Research ethical approval nos. 16/A019 and 18/A064), following written, informed consent. Patients had a median age of 70 years (range 29–95) and 43 of 78 were male.

**Gene expression quantification.** Total RNA from 245 individuals was quantified using Illumina HumanHT-12 v4 BeadChip gene expression arrays including 47,231 probes. Probe sequences mapping to more than one genomic locus, and probe sequences containing common genomic variation (minor allele frequency > 1%) were excluded from further analysis. Gene expression estimates were normalised (random-spline normalisation) before variance-stabilising transformation using the R package lumi[77]. Samples were processed in 2 batches, and gene expression estimates corrected using the the R package ComBat[78]. Following quality control, gene expression quantified at 29,011 probes, mapping to 18,078 unique genes were included in the analysis.

**Genotyping and imputation.** Genomic DNA was extracted from whole blood using Gentra Puregene Blood Kits (Qiagen) according to manufacturer's instructions, before dsDNA quantification using PicoGreen (Invitrogen). Genome-wide genotypes at 733,202 loci were generated using HumanOmniExpress-12v1.0 BeadChips (Illumina). SNP quality control (QC) filters were applied as follows: minor allele frequency (MAF) < 4%, Hardy-Weinberg equilibrium (HWE) $p < 1 \times 10^{-6}$, plate effect $p < 1 \times 10^{-6}$ and SNP missingness > 2%. We calculated identity by descent and principal components of LD-pruned genome-wide genotyping data to identify related individuals and sample outliers respectively. Samples with call rates < 98% were excluded from analysis. QC metrics were calculated in PLINK.

Following QC, 609,704 variants were taken forward for genome-wide imputation with pre-phasing using SHAPEIT[79] and imputation using IMPUTE2[80]. We used 1000G Phase 1 as a reference panel. Following imputation, SNPs with imputation info scores < 0.9 or HWE $p < 1 \times 10^{-6}$ were excluded from further analysis. Genotypes at 6,012,996 loci were taken forward for eQTL mapping. Principal components analysis of genome-wide genotyping data did not identify population outliers. Comparison of these genetic principal components with those of 1000G project reference samples confirmed European ancestry in all study samples (Supplementary Fig. 9). Genotyping QC did not identify related samples. All 245 individuals were included in the analysis.

For patients with metastatic melanoma, following genomic DNA extracted as described above, samples were genotyped using the Infinium Global Screening Array 24v3 (Illumina). Following QC (as above), 297,971 SNPs were taken forward for phasing and genome-wide imputation using Haplotype Reference Consortium v1[81] as a reference panel. Phasing was performed with Eagle2[82] and imputation with Minimac4[83], as implemented in the Michigan Imputation Server v1.2.4[83]. Following imputation, we retained SNPs with MAF > 0.04, R2 > 0.7 and HWE $p > 1 \times 10^{-6}$ for downstream analysis.

We used SHAPEIT phased genotypes at 100 SNPs in the KIR region on chromosome 19 to perform KIR copy number imputation using KIR*IMP v1.2.0[46]. KIR*IMP uses a bespoke reference panel of genotype and KIR copy number from individuals of European ancestry from the UK (698 individuals from 348 families). To assess the performance of KIR copy number imputation, we assayed the presence or absence of 12 functional KIR genes (2DL1-5, 2DS1-5, 3DL1, 3DS1) and two pseudogenes (2DP1 and 3DP1) in a subset of the study samples ($n = 171$). For each gene/pseudogene we used sequence-specific primers to amplify PCR products from genomic DNA. PCR products were separated with agarose gel electrophoresis and stained with ethidium bromide. The presence or absence of each KIR gene/pseudogene was confirmed with two complementary PCR reactions, resulting in two PCR products of different length at each locus.

**eQTL analysis.** QTLtools[10] was used for eQTL mapping under an additive linear model, including PCs of gene expression data to limit the effect of confounding variation. *cis* eQTL were defined as regulatory variation within 1Mb of the associated probe and *trans* loci those > 5Mb from the associated probe. Inclusion of 32 and 12 PCs in the eQTL analysis maximised eQTL discovery in *cis* and in *trans* respectively (Supplementary Fig. 1). Prior to eQTL mapping, expression data were corrected for PCs before rank normal transformation. For *cis* eQTL mapping, QTLtools approximates a permutation test at each phenotype, controlling for multiple-testing burden at the level of each phenotype. We used 10,000 permutations for *cis* eQTL mapping. To account for cases in which multiple probes mapped to the same gene we retained the probe with most significant, nominal *cis* association, as implemented with the --grp--best option in QTLtools. A second level of multiple-testing correction across all phenotypes tested was applied in R using qvalue[84]. Forward-backward stepwise regression implemented in QTLtools[10] was used to define multiple independent signals reaching a permutation-based significance threshold, mapping lead eSNPs at each signal to define identify secondary, tertiary and quaternary *cis* eQTL. For all analyses, we considered an $FDR < 0.05$ to be significant.

We mapped *trans* eQTL using QTLtools in two stages, firstly fitting additive linear models for each phenotype:genotype pair generating nominal $p$ values for each association, before determining the significance threshold by permuting phenotypes 1000 times. Correlation between phenotypes (but not genotypes) is maintained within each permutation run. By analogy to our approach for *cis* mapping, for cases in which multiple probes mapped to the same gene we retained the probe with the single most significant *trans* association for each gene. To alleviate false-positive associations in our *trans* analysis driven by SNPs in repetitive regions[85], we excluded SNPs falling within repeat regions as defined by the UCSC RepeatMasker track[86], taking forward 2,853,403 SNPs for analysis. False-positive *trans* associations can also be secondary to cross-mapping artefacts, whereby an array probe maps to paralogous genes mis-identifying a *cis* association as one acting in *trans* for a distal paralogous gene. To address this, following mapping in *trans* for each gene for which we identified a *trans* eQTL we used Re-Annotator[87] to re-map the array probe to a 10 Mb region centred on the identified *trans* eSNP. Allowing up to 6 mismatches, this re-mapping procedure identified 14 potential false-positive *trans*-eQTLs, which were excluded from further analysis. For significant *trans* associations, we grouped eSNPs located within a window of 1Mb, affecting the expression of a single gene, as potentially belonging to a single *trans* signal. We then tested whether each of these *trans* signals represented a single association, recapitulating mapping in *trans* for that probe conditioning on the peak eSNP.

**Identification of NK cell-specific eQTL.** To determine sharing of NK cell *cis* eQTL with other cells, moloc[11] was used in R to compare association at NK cell *cis* eQTL between NK cells and four other previously published datasets of eQTL in primary immune cell subsets from individuals of European ancestry[12–14]. Moloc adopts a Bayesian framework to compare models of association across multiple traits at a given genetic locus, using summary statistics. We applied moloc at all identified *cis* eQTL in NK cells, defining the evidence for shared regulatory effects on gene expression across NK cells (this dataset, $n = 245$), CD4$^+$ T cells ($n = 282$), CD8$^+$ T cells ($n = 271$), monocytes ($n = 414$) and neutrophils ($n = 101$). We considered an eQTL to be unique to NK cells where the posterior probability of the eQTL not being shared with another cell type > 0.8, and where a unique eQTL in NK cells was the most likely model overall (either an eQTL in NK cells is the only eQTL at that locus, or where other eQTL at the same locus are distinct from the eQTL in NK cells).

**Allele-specific expression.** To validate the effect of regulatory variation at *MC1R*, we used the C-BASE allele-specific expression assay[33,34], designing primers to full-length *MC1R* and cloning PCR product of cDNA and genomic from NK cells from 5 heterozygous individuals heterozygous using TA Cloning kit (Thermofisher). In addition, the same approach was used from monocyte cDNA samples for a separate 5 heterozygote individuals. Cloning product was transformed and colonies isolated with Taqman genotyping being performed on isolated colonies (spotted with pipette tip into PCR solution) using a probe designed to detect both alleles. Ninety-six colonies were tested for each sample, significance of effect for NK cells was performed using a ChiSquared test (1df), comparing ratios of genomic to cDNA. A paired $T$ test was performed for the monocyte samples.

**Colocalisation and enrichment.** We used ENCODE functional annotations[88] to interrogate colocalisation of chromatin states and transcription factor binding sites in lymphoblastoid cell lines (LCLs - GM12878) with eQTL in NK cells. We used ChIP-seq defined transcription factor binding sites for 52 transcription factors, and segmentation of the LCL genome into 7 functional tracks defined by ChromHMM[89] and SegWay[90]; promoter regions, promoter flanking regions, enhancers, weak enhancers or open chromatin *cis*-regulatory elements, CTCF enriched elements, transcribed regions, and repressed or low activity regions. We calculated the density of functional annotations surrounding each *cis* eSNP, counting the number of annotations falling into 1kb bins up to 1Mb from each feature. We used a permutation approach implemented in QTLtools to calculate

evidence for enrichment of eSNPs with functional annotations, calculating the frequency of observed overlap between a given functional annotation and an eSNP, comparing this to the number of overlaps expected by chance (permuting phenotypes across all probes tested). We tested for enriched biological pathways among *cis* NK cell-specific eQTL using the GOBP database in XGR[19]. Genes with *cis* NK cell-specific eQTL were tested against a background of all genes tested in the eQTL analysis, using a hypergeometric test.

We used two approaches to test for colocalisation of causal variants between *cis* eQTL and GWAS traits. Firstly we used Regulatory Trait Concordance (RTC)[20] implemented in QTLtools. RTC integrates GWAS hits, eQTL data and local LD structure to quantify the decrease in eQTL significance when the phenotype is residualised for the GWAS hit. We used RTC to test for colocalisation at *cis* eQTL with 87,919 GWAS-significant loci ($p < 5 \times 10^{-8}$) from 5,823 GWAS studies/traits downloaded on 28/01/2021 from the NHGRI-EBI GWAS Catalog[8] (Supplementary Data 16). An advantage of the RTC method is that it only requires the identity of GWAS-significant peak SNPs, rather than complete GWAS summary statistics. We considered RTC scores > 0.9 to be significant. Secondly, we used the R package coloc v5.1.0[21] to identify evidence of causal variants shared by NK cell eQTL and GWAS loci. Coloc adopts a Bayesian approach to compare evidence for independent or shared association signals for two traits at a given genetic locus. We tested for colocalisation between NK cell primary *cis* eQTL and 100 GWAS traits (Supplementary Data 7) with evidence of trait association ($p < \times 10^{-6}$) within a 250kb window of the peak eSNP at each significant NK cell primary *cis* eQTL. We used the coloc.signals function[91] to allow for multiple, independent causal signals in both traits. We considered a posterior probability > 0.8 supporting a shared causal locus to be significant.

To test for enrichment of colocalisation of GWAS traits among NK cell eQTL, we used Fisher exact tests to compare the proportion of NK cell eQTL within 250 kb of a GWAS locus with evidence of trait association ($p < \times 10^{-6}$) for which their is evidence of a shared causal locus, comparing this to the proportion observed for a null GWAS trait (height). We considered traits for which there were at least 5 NK cell eQTL within 250 kb of a GWAS locus. Throughout we applied FDR correction to account for the number of annotations and traits tested.

**TWAS analysis**. We performed transcriptome-wide association studies (TWAS) implemented in FUSION[92] of five autoimmune diseases, using functional gene expression weights calculated from the NK cell eQTL data described here to impute NK cell gene expression into previously published GWAS of ulcerative colitis[50], Crohn's disease[50], systemic lupus erythematosus[52], primary biliary cirrhosis[53] and rheumatoid arthritis[51]. For each gene, we correct normalised gene expression for PCs of gene expression ($n = 32$ as for eQTL mapping) and genome-wide genotypes ($n = 10$) and estimate *cis* genetic heritability (SNPs within 1Mb of the TSS), retaining significantly heritable genes ($p < 0.05$). For heritable genes we cross-validated genotype-imputed gene expression estimates, selecting the best-performing model (highest $R^2$) from 5 prediction models; best linear unbiased predictor, Bayesian linear mixed model, elastic-net regression ($\alpha = 0.5$), LASSO regression and single best eQTL. We then pass functional weights from the best-performing of these models and GWAS summary statistics to FUSION to compute TWAS association statistics for each gene. We applied FDR correction across all genes included in each TWAS, considering FDR < 0.05 to be significant. To address whether trait-associated genes were conditionally independent, we performed conditional stepwise regression including all significant TWAS genes on a single chromosome. Finally, to complement the TWAS analysis, at each TWAS-significant gene we used coloc to assess the probability that the GWAS risk locus and NK cell eQTL share a causal variant.

**Multiparametric flow cytometry**. Whole blood from each volunteer was collected into EDTA-containing collection tubes (Vacutainer System, Becton Dickinson). PBMCs were purified by Ficoll-Pacque density gradient centrifugation, before storage at −80 °C. Frozen PBMCs were thawed, counted and plated in 96-well v-bottomed polystyrene plates (Corning, USA) at a concentration of $10^6$ cells per well. After staining with viability dye (LIVE/DEAD FixableNear-IR Dead Cell Stain, cat L10119, Thermo-Fisher Scientific) at a concentration of 1:1000 in PBS, cells were surface immunostained with BUV496-conjugated anti-CD3 (1:150 dilution, clone UCHT1, cat 612940, BD Biosciences), BUV395-conjugated anti-CD56 (1:50 dilution, clone NCAM16.2, cat 563555, BD Biosciences), VioGreen-conjugated PanKIR2D (1:50 dilution, clone 130-128-216, Miltenyi Biotec), BV785-conjugated anti-CD57 (1:50 dilution, clone QA17A04, cat 393329, Biolegend) and PE-conjugated anti-CD226 (1:100 dilution, clone 11A8, cat 338305, Biolegend) in HBSS supplemented with 5% FBS. Immunostaining was performed in the dark, on ice for 30 min. Flow cytometry was performed using a BD LSR Fortessa X20 flow cytometer, and the data analysed (Supplementary Fig. 10) using FlowJo v10.8 Software (BD Life Sciences). Proportions of CD57$^+$, CD226$^+$ and KIR2D$^+$ NK cells were logit transformed prior to analysis, and genotype-phenotype associations tested with linear regression. *P* values are calculated with likelihood ratio tests.

**Reporting summary**. Further information on research design is available in the Nature Research Reporting Summary linked to this article.

## Data availability

Sample genotypes are available at the European Genome-Phenome Archive (EGA) with accession ID EGAS00000000109. Raw gene expression data and probe QC filters have been deposited at: https://doi.org/10.5281/zenodo.6352656. Summary statistics are available via R Shiny, interactive, browser-based applications for *cis* mapping data (https://jjgilchrist.shinyapps.io/nk_cis_eqtl/) and for *trans*-mapping data (https://jjgilchrist.shinyapps.io/nk_trans_eqtl/). We have also contributed raw genotype and phenotype data to the eQTL Catalogue project (https://www.ebi.ac.uk/eqtl/).

The study makes use of the following publicly-available data; Gene Ontology consortium (http://www.geneontology.org), ENCODE Project (https://www.encodeproject.org), Open Targets Genetics (https://genetics.opentargets.org). UK Biobank summary statistics (http://www.nealelab.is/uk-biobank/), eQTLGEN Consortium (https://www.eqtlgen.org). Source data are provided with this paper.

## Code availability

Scripts and source data used to produce figures, alongside scripts used in data analysis are available at: https://github.com/jjgilchrist/NK_eQTL[93].

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

## Acknowledgements

J.C.K. is supported by Wellcome Trust Investigator Award [204969/Z/16/Z], NIHR Oxford Biomedical Research Centre and Chinese Academy of Medical Sciences (CAMS) Innovation 537 Fund for Medical Science (grant number: 2018-I2M-2-002), Wellcome Trust Grants 090532/Z/09/Z and 203141/Z/16/Z to core facilities Wellcome Centre for Human Genetics, Oxford Biomedical Research Computing (BMRC) facility, a joint development between the Wellcome Centre for Human Genetics and the Big Data Institute supported by Health Data Research UK and the NIHR Oxford Biomedical Research Centre. The study was funded by Wellcome Trust Intermediate Clinical Fellowship to B.P.F. (no. 201488/Z/16/Z). J.J.G. is funded by a National Institute for Health Research (NIHR) Clinical Lectureship. The views expressed are those of the author(s) and not necessarily those of the NHS, the NIHR or the Department of Health and Social Care.

## Author contributions

Author contributions were as follows: J.J.G., W.L., S.K., O.T., E.N. and B.P.F. performed the statistical and computational analysis. S.M., V.N., E.L., S.D. and B.P.F. recruited the eQTL study samples, isolated NK cells and performed RNA extraction. P.K.S., S.K., O.T., C.A.T., R.A.W., A.V.A., R.C. and B.P.F. recruited melanoma patients, isolated NK cells and performed flow cytometry. D.H-B. and J.W. performed the PCR validation of imputed KIR types. J.J.G., and B.P.F. wrote the manuscript. The study was designed and managed by J.C.K. and B.P.F.

## Competing interests

The authors declare no competing interests.
