## [Peer Review File · Nature Communications]

Natural Killer cells demonstrate distinct eQTL and transcriptome-wide disease associations, highlighting their role in autoimmunityREVIEWER COMMENTS

Reviewer #1 (Remarks to the Author):

Gilchrist et al. Have performed a well-powered eQTL study in an specific innate immunity cell type, Natural-Killer cells. Their results are relevant and support an important role of NK-cells in autoimmune disorders. The manuscript is well-written and clear. Moreover, the selected methodology is appropriated for the study aims and the conclusions are concordant with the observed data. Nevertheless, there are several concerns that should be addressed before considering this manuscript for publication:

Major points:

1. The transcriptome data included in this report was not obtained using the RNA-seq technology. Although the use of expression arrays surely increased the number of individuals that were included in this study. I consider that the authors should acknowledge the limitations of this method in the discussion section.
2. The imputation of the genotype data was performed using the 1000 Genomes Project a reference panel while the imputation of KIR was performed using the UK Biobank. The authors should justify this discrepancy.
3. Statistical power should be reported for the identification of both cis and trans eQTLs.
4. The number of permutations performed for p-value correction is very limited. I would strongly recommend to perform at least 1000 permutations.
5. The authors state that "Much of trans -acting regulatory variation is secondary to cis - effects on upstream regulatory genes". However, less than 20% of the described trans-eQTLs have cis-mediation. The context of these figures should be explained.
6. The relation between pigmentation and immunity reported for TMEM163 should be further described as this connection seems difficult to explain.
7. Height was selected as background for eQTL enrichment for other traits, do the results change if a different trait is used as a background?
8. How many TWAS gene-trait pairs are NK-specific?
9. Do NK-specific eQTLs correspond to highly expressed genes in this cell type?
10. It would really help to increase data-mining and impact if a link or web browser access to the eQTL results were provided.

Minor points:

1. Figure S6 is not clear for the reader and hard to interpret.
2. Principal component analysis using genotype data and reference populations should be shown to confirm the ancestry of the studied individuals.
3. QC data should be provided for the expression arrays.
4. Sentences such as "32 and 20 PCs maximised eQTL discovery" are not clear and should be modified.

Reviewer #2 (Remarks to the Author):

Manuscript 311469: "Natural Killer cells demonstrate distinct eQTL and transcriptome-wide disease associations, highlighting their role in autoimmunity"

In this manuscript, the authors measured gene expression from isolated NK cells from 245 donors with genotypes to perform cis and trans eQTLs analysis to identify cell-specific genetic regulation of gene expression. Co-localization and TWAS analysis identify candidate genes mediating complex diseases via expression regulation in NK cells.

The manuscript is well written with methods and objective clearly define. In general the findings here reported provide an important resource of cell-specific genetic regulation and candidate genes with relevance to understand the genetics of complex diseases involving the immune system or

inflammation processes. With the exception of the trans-QTL analysis and some missing information I find the study was properly performed. Some specific comments follow.

1) The trans-eQTL analysis reported ~5700 trans-eSNPs for 116 genes with a sample size of 245. This is orders of magnitude larger than studies with larger sample size have previously reported and their results are hard to believe. It is possible that the authors are reporting all signals below their threshold, and not the independent signals, so please clarify this point. If the authors included all SNPs, please report independent signals or loci defined by other means if the calculations are not possible.

An issue not considered also is the known higher rate of false positives produced by trans-QTL analyses. One of the known issues comes from SNPs within repeat regions or of low complexity, as those may generate false positives with nearby cis-eQTLs. Removing SNPs within regions identified by RepeatMarker or confirming those were not included is therefore recommended. For details the authors may look at Saha & Battle, F1000, 2019. Although the referenced paper mostly discusses RNAseq issues not applicable to this manuscript, the reviews, references and discussion may be useful. In addition, a supplemental note in Vösa et al (BioRxiv 447367) provides also some additional information about QC considerations. The authors should provide evidence that they have considered sources of false positives here and that their associations are genuine.

The authors did not attempt to perform any replication of these signals. I understand it is not possible to find other expression datasets of NK cells with genotypes, however there are trans-eQTL analyses from whole blood available, which include NK cells. As it is possible that not all signals identified in this study are cell-specific, and some are also present in bulk blood studies. To make it clear: lack of replication should NOT be taken as a failure of the study, but as a complementary analysis associated to the cell-specific aspect of this study and to strengthen their findings.

2) The authors evaluated after quality check ~29k probes, but ~18k genes. How were probes from the same gene handled? In case authors did not control for correlated phenotypes, the option "--grp-best" from QTLtools can be used. If they have, please indicate this in the methods. This affects the trans-QTL analysis too, more even if one considered the higher false positive rates that may arise from the inclusion of highly correlated phenotypes treated independently.

3) The authors seem to distinguish between cis-eQTLs discovered in the 1st, 2nd, 3rd or 4th iteration of the conditional analysis. With no additional information, I understand this refers to the order of discovery, which lacks any biological meaning. The order of discovery is influenced by technical factors such as imputation accuracy and imperfect LD which determine what signals are first identified and all subsequent. What's the point of making the distinction in some points of the manuscript between 2nd, 3rd, or 4th eSNP? Are the authors learning something here giving the lack of biological meaning in the order?

Furthermore, on page 5 the authors investigate the cis-eQTLs functions and report among other analyses enrichment of secondary eQTLs for enhancers. Shall we assume this category includes only cis-eSNPs discovered during the second iteration or all eSNPs discovered after the first iteration?

4) Table S1, with the RTC results does not include the output of the analysis. I was expecting to be able to evaluate the distribution of RTC scores, R² of co-localizing SNPs and other metrics provided by RTC.

5) It is not clear what version of COLOC the authors have used, and I cannot tell if the model used assumes a single eSNP effect per gene-trait, the only option in older versions, or if they have accounted for multiple genetic signals as newer versions allow. If the authors have not accounted for secondary eQTLs, they should re-run their analysis with the new version of COLOC, or produce pseudo-phenotypes for expression that control for secondary eSNPs to avoid false positives due to tagging signals. If this has been done, please make it clear in the methods to avoid confusion.

6) The authors used the 1000 Genomes Phase 1 panel for imputation, while newer versions and other panels have been available from more than 5 years. It is often hard to keep up with new developments and updates, and I am not asking the authors to re-do their entire analysis with a new panel, but I would like to understand the situation that makes them use such an old panel, whether this is because the imputation was already available, or the project has taken >5 years from starting to completion.

Reviewer #3 (Remarks to the Author):

Gilchrist and colleagues have performed an elegant analysis of eQTLs to profile eQTLs seemingly specific to NK cells that are linked to diseases, such as SLE, UC, Chron's, etc. The authors used RNA expression data and performed a transcriptome-wide association study across several autoimmunity diseases. They identified 27 genes with novel associations. The authors' analyses revealed both cis and trans regulatory elements that may provide future benefit to target in the clinic.

I believe this study has potential to be informative to Nature Communication's readership. However, it lacks any functional validation data. This study lacks any immune profiling at the protein level looking at causes or effects for NK cell functional attributes comparing healthy and diseased individuals. Whether in blood or in relevant tissues, I would like to see at a minimum staining for DNAM-1/CD226 along with some of the others defined for this analysis.

Without corroborative evidence showing effects on NK cell protein-level phenotypes and functional consequences, I do not feel that this manuscript is appropriate for Nature Communications. I would be happy to review any modified manuscript from this group.

Reviewer #1

Gilchrist et al. Have performed a well-powered eQTL study in an specific innate immunity cell type, Natural-Killer cells. Their results are relevant and support an important role of NK-cells in autoimmune disorders. The manuscript is well-written and clear. Moreover, the selected methodology is appropriated for the study aims and the conclusions are concordant with the observed data. Nevertheless, there are several concerns that should be addressed before considering this manuscript for publication:

Major points:

1. The transcriptome data included in this report was not obtained using the RNA-seq technology. Although the use of expression arrays surely increased the number of individuals that were included in this study. I consider that the authors should acknowledge the limitations of this method in the discussion section.

Many thanks. We have added the following to the discussion to reflect this (lines 332-337):

“Our study has some limitations. Most notably, the expression data we present here was generated with microarray technology rather than with RNA sequencing. Large-scale meta-analysis of eQTL mapping in whole blood have demonstrated excellent concordance between studies performed with microarray and RNAseq. However, within this dataset we are unable to identify genetic mediators of expression in NK cells for transcripts lacking probe coverage, alternative splicing and exon/transcript usage.”

2. The imputation of the genotype data was performed using the 1000 Genomes Project a reference panel while the imputation of KIR was performed using the UK Biobank. The authors should justify this discrepancy.

Apologies that this was not adequately clear. KIR imputation was performed with KIR*IMP which uses a bespoke imputation reference panel of individuals of European ancestry. We have added the following to the text to clarify this (lines 391-393): “KIR*IMP uses a bespoke reference panel of genotype and KIR copy number from individuals of European ancestry from the UK (698 individuals from 348 families).”

3. Statistical power should be reported for the identification of both cis and trans eQTLs.

Many thanks for this suggestion. We have included estimated power for cis and trans variant discovery within our data in a new Supplementary Figure (Supplementary Fig. 8). We have included the following text in the manuscript (lines 353-363): “We estimated study power to detect trait-associated genetic variation in cis and trans across a range of minor allele frequencies and numbers of independent tests (Supplementary Fig. 8). In each case we used powerEQTL to calculate power for $\alpha=0.05/\text{number of tests}$ and an effect size and standard deviation of 0.13. For cis power calculations we assume a variable number of independent tests, defined as the number of independent SNPs in a cis testing window (1-5,000) multiplied by the number of genes tested (18,000). For trans power calculations we assume 18×10^9 independent tests, assuming 18,000 genes and 1×10^6 independent SNPs genome-wide.

Assuming an effective sample size of 245, our study has 80% power to detect a cis eQTL with a causal eSNP with minor allele frequency >0.08, assuming 100 independent tests in the cis testing window. In trans, our study has 80% power to detect a trans eQTL with a causal eSNP with minor allele frequency >0.12.”

4. The number of permutations performed for p-value correction is very limited. I would strongly recommend to perform at least 1000 permutations.

Many thanks for highlighting this. In the cis analysis we had permuted 10,000 times, and we have now made this clear in the text (line 408). As suggested, we have increased the number of permutations used in the revised trans analysis to 1,000 (line 418).

5. The authors state that “Much of trans -acting regulatory variation is secondary to cis - effects on upstream regulatory genes”. However, less than 20% of the described trans-eQTLs have cis-mediation. The context of these figures should be explained.

Many thanks for highlighting that we overstated this. We have modified the text to read (lines 172-174): “Regulatory variation acting in trans is enriched for cis-eSNPs, suggesting that a proportion of trans-acting variation operates via cis effects on upstream regulatory genes.”

6. The relation between pigmentation and immunity reported for TMEM163 should be further described as this connection seems difficult to explain.

Many thanks for asking us to expand on this. We have added the following to the text (lines 296-301): “Immunodeficiencies with combined NK cell dysfunction and disorders of pigmentation, including Chediak-Higashi, Hermansky-Pudlak and Griscelli syndromes, are characterised by impairment of secretory lysosome function. Given the described role of TMEM163 in lysosome composition, it is tempting to speculate that TMEM163 may modify both pigmentation and risk of immune-mediated disease modifies through effects on melanin secretion and NK cell cytotoxic degranulation.”

7. Height was selected as background for eQTL enrichment for other traits, do the results change if a different trait is used as a background?

Many thanks for this suggestion. We now recapitulate our enrichment analysis using an alternative background (whole body impedance: UK Biobank phenotype code: 23106_irnt). We describe this in the text as follows (lines 136-141): “To investigate whether the choice of height as background for enrichment could bias our results, we recapitulated our analysis using whole body impedance (UK Biobank phenotype code: 23106_irnt) as an alternative background. In that analysis we find no evidence that choice of background significantly affects our results, with our enrichment estimates derived using height as background being well-correlated ($r=0.575$, $p=5.51e-06$) with those using whole body impedance (Supplementary Fig. 5)”

8. How many TWAS gene-trait pairs are NK-specific?

Many thanks for highlighting that we had not included this information. We now include the following in the text (lines 269-276): “We then sought to define

which of these gene-trait pairs are explained by an NK cell-specific cis eQTL, that is instances in which the most likely model includes an eQTL signal unique to NK cells (see Methods) and the eQTL in NK cells colocalises with the GWAS signal ($PP>0.8$). In that analysis, 6 gene-traits pairs are driven by gene expression in NK cells specifically; CD226 expression modifies systemic lupus erythematosus and primary biliary cirrhosis risk, TMEM163 expression modifies primary biliary cirrhosis risk, CALHM6 expression modifies ulcerative colitis risk, and both ZMIZ1 and OSM modify Crohn's disease risk.”.

9. Do NK-specific eQTLs correspond to highly expressed genes in this cell type?

Many thanks for this suggestion. We now include these data in a new Supplementary Fig. 2. Interestingly there is no correlation. We now include the following in the text (lines 72-73): “there is no significant difference ($p=0.152$) in median expression of genes with shared or NK cell-specific eQTL (Supplementary Fig. 2).”

10. It would really help to increase data-mining and impact if a link or web browser access to the eQTL results were provided.

We very much agree. We have added the following as a Data Availability statement: Summary statistics are available via R Shiny, interactive, browser-based applications for cis mapping data (https://jgilchrist.shinyapps.io/nk_cis_eqtl/) and for trans mapping data (https://jgilchrist.shinyapps.io/nk_trans_eqtl/). We have also contributed raw genotype and phenotype data to the eQTL Catalogue project (<https://www.ebi.ac.uk/eqtl/>).

Minor points:

1. Figure S6 is not clear for the reader and hard to interpret.

Many thanks. On reflection we feel that this added little to the manuscript and have removed it in its entirety.

2. Principal component analysis using genotype data and reference populations should be shown to confirm the ancestry of the studied individuals.

Many thanks. We now include a supplementary figure (Supplementary Fig. 9) plotting the first two principal components of genome-wide genotyping data for our study samples, comparing them to 1000G project reference samples. This confirms European ancestry for all 245 study samples.

3. QC data should be provided for the expression arrays.

Many thanks. We now provide expression array QC data with the deposited raw array expression data at: [10.5281/zenodo.6352656](https://zenodo.org/record/6352656).

4. Sentences such as “32 and 20 PCs maximised eQTL discovery” are not clear and should be modified.

Many thanks we have tried to improve the clarity of the language throughout.

Reviewer #2

Manuscript 311469: “Natural Killer cells demonstrate distinct eQTL and transcriptome-wide disease associations, highlighting their role in autoimmunity”

In this manuscript, the authors measured gene expression from isolated NK cells from 245 donors with genotypes to perform cis and trans eQTLs analysis to identify cell-specific genetic regulation of gene expression. Co-localization and TWAS analysis identify candidate genes mediating complex diseases via expression regulation in NK cells.

The manuscript is well written with methods and objective clearly define. In general the findings here reported provide an important resource of cell-specific genetic regulation and candidate genes with relevance to understand the genetics of complex diseases involving the immune system or inflammation processes. With the exception of the trans-QTL analysis and some missing information I find the study was properly performed. Some specific comments follow.

1) The trans-eQTL analysis reported ~5700 trans-eSNPs for 116 genes with a sample size of 245. This is orders of magnitude larger that studies with larger sample size have previously reported and their results are hard to believe. It is possible that the authors are reporting all signals bellow their threshold, and not the independent signals, so please clarify this point. If the authors included all SNPs, please report independent signals or loci defined by other means if the calculations are not possible.

Many thanks for highlighting that was not adequately clear. Yes, the 5,700 trans-eSNPs referred to the total (not independent) signals. Having modified

the trans analysis (as suggested below), we now report a total of 2,266 SNPs with significant trans effects at 84 independent loci. We describe how we defined independent loci in the text as follows (lines 430-434): “For significant trans associations, we grouped eSNPs located within a window of 1Mb, affecting the expression of a single gene, as potentially belonging to a single trans signal. We then tested whether each of these trans signals represented a single association, recapitulating mapping in trans for that probe conditioning on the peak eSNP.”

An issue not considered also is the known higher rate of false positives produced by trans-QTL analyses. One of the known issues comes from SNPs within repeat regions or of low complexity, as those may generate false positives with nearby cis-eQTLs. Removing SNPs within regions identified by RepeatMarker or confirming those were not included is therefore recommended. For details the authors may look at Saha & Battle, F1000, 2019. Although the referenced paper mostly discusses RNAseq issues not applicable to this manuscript, the reviews, references and discussion may be useful. In addition, a supplemental note in Võsa et al (BioRxiv 447367) provides also some additional information about QC considerations. The authors should provide evidence that they have considered sources of false positives here and that their associations are genuine.

Very many thanks for these important points. We have revised our trans analysis to better account for false positive discovery. We describe this in the methods as follows (lines: 421-430): “To alleviate false positive associations in our trans analysis driven by SNPs in repetitive regions, we excluded SNPs falling within repeat regions as defined by the UCSC RepeatMasker track,

taking forward 2,853,403 SNPs for analysis. False positive trans associations can also be secondary to cross-mapping artefacts, whereby an array probe maps to paralogous genes mis-identifying a cis association as one acting in trans for a distal paralogous gene. To address this, following mapping in trans, for each gene for which we identified a trans eQTL we used Re-Annotator to re-map the array probe to a 10Mb region centred on the identified trans eSNP. Allowing up to 6 mismatches, this re-mapping procedure identified 14 potential false positive trans eQTLs, which were excluded from further analysis.”

The authors did not attempt to perform any replication of these signals. I understand It is not possible to find other expression datasets of NK cells with genotypes, however there are trans-eQTL analysis from whole blood available, which include NK cells. As it is possible that not all signals identify in this study are cell-specific, and some are also present in bulk blood studies. To make it clear: lack of replication should NOT be taken as a failure of the study, but as a complementary analysis associated to the cell-specific aspect of this study and to strength their findings.

Many thanks for raising this important point. To address this, we have investigated replication of our NK cell trans associations in the eQTLGen dataset. As you suggest, our expectation was that our ability to replicate in whole blood would be limited, and while this is clearly the case, we do find some informative instances of replication of NK cell trans eQTL in the eTLGen data. We set this out in the text as follows (lines:239-259): “We further sought to replicate evidence for our trans associations in NK cells in the large-scale meta-analysis of whole blood eQTL data compiled by the eQTLGEN consortium. NK cells represent a small proportion of the circulating leukocytes

in whole blood. Given that trans eQTL hubs are frequently thought to be cell-type specific, our expectation was that our power to replicate trans associations in whole blood eQTL would be limited. Despite this, we were able to replicate a small number of NK cell trans associations observed in our study in the eQTLGEN consortium data (Supplementary Table 8). We find evidence of replication for the effect of rs3811444 on JAM3 expression and for rs10876864 on KCTD11 expression in whole blood (Supplementary Table 8). In both cases the replicated trans association forms part of a larger trans-regulated gene network in whole blood, with rs3811444 regulating the expression of 97 genes in trans and rs10876864 the expression of 47. Moreover, while there is no direct evidence for replication of the trans network mediated by MC1R expression, rs9939914 (a significant trans eSNP for both DRD3 and SNORD85 in our data) affects the expression of a network of four genes in trans in whole blood; GPR25, PHF17, LDHD, BBS10. These observations are in keeping with a model in which, while some trans regulatory eSNPs are likely to be shared across cell type and context, the gene network which they regulate will vary according to cell type. In keeping with this, in the case of the trans regulatory hub including rs10876864, for which we see evidence of replication of its effect on KCTD11 expression in whole blood, we have previously reported trans effects at this locus on LAP3P2, IP6K2 and HELZ2 in B cells and LAP3P2 in monocytes.”

2) The authors evaluated after quality check ~29k probes, but ~18k genes. How were probes from the same gene handled? In case authors did not control for correlated phenotypes, the option "--grp-best" from QTLtools can be used. If they

have, please indicate this in the methods. This affect the trans-QTL analysis too, more even if one considered the higher false positives rates that may raise from the inclusion of highly correlated phenotypes treated independently.

Many thanks for highlighting that this was not adequately clear. We did control for correlated phenotypes (probes mapping to the same gene), selecting the probe for each gene with the most significant, nominal cis association, as implemented with the "--grp-best" option in QTLtools. For our trans analysis we adopted an analogous approach, selecting the probe for each gene with the most significant, nominal trans association. We have clarified this in the methods (line 410).

3) The authors seem to distinguish between cis-eQTLs discovered in the 1st, 2nd, 3rd or 4th iteration of the conditional analysis. With no additional information, I understand this refers to the order of discovery, which lacks any biological meaning. The order of discovery is influenced by technical factors such as imputation accuracy and imperfect LD which determine what signals if first identified and all subsequent. What's the point of making the distinction in some points of the manuscript between 2nd, 3rd, or 4th eSNP? Are the authors learning something here giving the lack of biological meaning in the order?

Many thanks for highlighting this. 1st, 2nd, 3rd order does indeed reflect order of discovery here. Our intention was not that the order of discovery was important, but rather that it was of interest that, given our sample size, up to 4 independent cis eQTL are observable for a minority of genes. We have amended the text to better reflect this (e.g. line 84).

Furthermore, on page 5 the authors investigate the cis-eQTLs functions and report among other analyses enrichment of secondary eQTLs for enhancers. Shall we assume this category includes only cis-eSNPs discovered during the second iteration or all eSNPs discovered after the first iteration?

In our initial analysis we had included only the second iteration of conditional analysis. We acknowledge that it makes more biological sense to perform this analysis including all eSNPs discovered following this first iteration, and have revised this analysis accordingly (see line 84).

4) Table S1, with the RTC results does not include the output of the analysis. I was expecting to be able to evaluate the distribution of RTC scores, R2 of co-localizing SNPs and other metrics provided by RTC.

Apologies for this oversight. RTC scores, SNP distance and LD information are now provided in a new supplementary table (Supplementary Table 5).

5) It is not clear what version of COLOC the authors have used, and I cannot tell if the model used assumes a single eSNP effect per gene-trait, the only option in older versions, or if they have accounted for multiple genetic signals as newer versions allow. If the authors have not accounted for secondary eQTLs, they should re-run their analysis with the new version of COLOC, or produce pseudo-phenotypes for expression that control for secondary eSNPs to avoid false positives due to tagging signals. If this has been done, please make it clear in the methods to avoid confusion.

Many thanks for raising this important point. We had used an older version of COLOC in our original analysis which did only assume a single signal per locus. We have recapitulated that analysis using COLOC v5.1.0, using the coloc.signals() function, allowing for multiple signals per trait. The amending manuscript (and supplementary table) reflect the results of this new analysis, which while different do not substantially alter our conclusion. We now detail this in the revised methods (lines: 484-485).

6) The authors used the 1000 Genomes Phase 1 panel for imputation, while newer versions and other panels have been available from more than 5 years. It is often hard to keep up with new developments and updates, and I am not asking the authors to re-do their entire analysis with a new panel, but I would like to understand the situation that make them use such an old panel, whether this is because the imputation was already available, or the project has taken >5years from starting to completion.

Many thanks for highlighting this. Imputed data for this cohort was generated concurrently with our data describing eQTL in monocytes and neutrophils. As we use the monocyte and neutrophil eQTL data here to consider evidence for shared/unique eQTL in NK cells, it facilitated cross comparison between studies to keep the reference panel consistent. Given that we focus on common genetic variation in a European population in this study, combined with stringent imputation QC, we feel that re-imputation with an updated reference panel would not materially change our results.

Reviewer #3

Gilchrist and colleagues have performed an elegant analysis of eQTLs to profile eQTLs seemingly specific to NK cells that are linked to diseases, such as SLE, UC, Chron's, etc. The authors used RNA expression data and performed a transcriptome-wide association study across several autoimmunity diseases. They identified 27 genes with novel associations. The authors' analyses revealed both cis and trans regulatory elements that may provide future benefit to target in the clinic.

I believe this study has potential to be informative to Nature Communication's readership. However, it lacks any functional validation data. This study lacks any immune profiling at the protein level looking at causes or effects for NK cell functional attributes comparing healthy and diseased individuals. Whether in blood or in relevant tissues, I would like to see at a minimum staining for DNAM-1/CD226 along with some of the others defined for this analysis.

Without corroborative evidence showing effects on NK cell protein-level phenotypes and functional consequences, I do not feel that this manuscript is appropriate for Nature Communications. I would be happy to review any modified manuscript from this group.

We are very grateful to the reviewer for their time to review our work and this helpful suggestion. Whilst there is good evidence to support disease risk alleles mediating their function at the level of transcript, we agree with the reviewer that protein level corroboration would be ideal.

We did not have access to the cells from this original cohort, thus sought to replicate three of our observed eQTL that were associated with surface proteins by using immunophenotyping of NK cells from a well-characterised patient cohort developed by the group. In PBMC samples from this cohort we explored the effects of eSNPs identified in our primary analysis on the circulating proportion of CD226⁺, CD57⁺ and KIR2D⁺ NK cells from untreated patients, with the experimenters blinded to patient genotype until all measurements had been taken. Given the added variation when dealing with patient derived samples and performing flow cytometry we were pleased to be able to replicate the genetic effects of the eSNPs on surface proteins for all three of these tested genes. In doing so we corroborated the functional importance of these observations in a disease relevant context. We set out these data in new results (“Protein phenotypes of NK cell eQTL”) and methods (“Multiparametric flow cytometry”) sections and new Figure 4.

REVIEWERS' COMMENTS

Reviewer #1 (Remarks to the Author):

The authors have properly addressed my previous concerns and the manuscript has improved remarkably after the revision. This reviewer has checked the availability of the data and the access through the shiny app and it worked satisfactorily.

Reviewer #2 (Remarks to the Author):

After evaluation, I consider the authors have adequately replied to all my comments. I want to thank the authors for making code and summary statistics available with their publication and for considering my comments.

I do not have any additional comments regarding methodology, but please consider incorporating some of the following suggestions:

- Line 209: the word "affecting" is written twice.
- Lines 239 to 259: It is not clear how replication was assessed, so please report somewhere the replication pvalue/multiple-testing and threshold used.

Reviewer #3 (Remarks to the Author):

Gilchrist and colleagues have done a nice job responding to all 3 reviewers' comments. I recognize my one sole criticism and request was not entirely straightforward logistically. But I feel that it definitely strengthens the authors' manuscript.

I believe the manuscript is well suited for Nature communications in its current revised form. Congrats to the co-authors.